# Pilot Study on the Impact of Polymorphisms Linked to Multi-Kinase Inhibitor Metabolism on Lenvatinib Side Effects in Patients with Advanced Thyroid Cancer

**DOI:** 10.3390/ijms24065496

**Published:** 2023-03-13

**Authors:** Silvia Cantara, Cristina Dalmiglio, Carlotta Marzocchi, Alfonso Sagnella, Lucia Brilli, Andrea Trimarchi, Fabio Maino, Laura Valerio, Maria Grazia Castagna

**Affiliations:** 1Department of Medical, Surgical and Neurological Sciences, University of Siena, 53100 Siena, Italy; 2Laboratory of Clinical and Translational Research, AOU Siena, 53100 Siena, Italy

**Keywords:** thyroid cancer, lenvatinib, multi-kinase inhibitors, SNPs, cytochrome P450, ATP-binding cassette transporters

## Abstract

Multi-kinase inhibitors (MKIs) represent the best therapeutic option in advanced thyroid cancer patients. The therapeutic efficacy and toxicity of MKIs are very heterogeneous and are difficult to predict before starting treatment. Moreover, due to the development of severe adverse events, it is necessary to interrupt the therapy some patients. Using a pharmacogenetic approach, we evaluated polymorphisms in genes coding for proteins involved with the absorption and elimination of the drug in 18 advanced thyroid cancer patients treated with lenvatinib, and correlated the genetic background with (1) diarrhea, nausea, vomiting and epigastric pain; (2) oral mucositis and xerostomia; (3) hypertension and proteinuria; (4) asthenia; (5) anorexia and weight loss; (6) hand foot syndrome. Analyzed variants belong to cytochrome P450 (*CYP3A4* rs2242480 and rs2687116 and *CYP3A5* rs776746) genes and to ATP-binding cassette transporters (*ABCB1* rs1045642, rs2032582 and rs2235048 and *ABCG2* rs2231142). Our results suggest that the GG genotype for rs2242480 in *CYP3A4* and CC genotype in rs776746 for *CYP3A5* were both associated with the presence of hypertension. Being heterozygous for SNPs in the *ABCB1* gene (rs1045642 and 2235048) implicated a higher grade of weight loss. The *ABCG2* rs2231142 statistically correlated with a higher extent of mucositis and xerostomia (CC genotype). Heterozygous and rare homozygous genotypes for rs2242480 in *CYP3A4* and for rs776746 for *CYP3A5* were found to be statistically linked to a worse outcome. Evaluating the genetic profile before starting lenvatinib treatment may help to predict the occurrence and grade of some side effects, and may contribute to improving patient management.

## 1. Introduction

Targeted therapy for the treatment of locally advanced or metastatic thyroid cancer has been introduced in recent decades. Nowadays, multi-kinase inhibitors (MKIs) represent the best therapeutic option in advanced thyroid cancer patients [1].

MKIs are a class of drugs that inhibit mitogenic signals via the phosphorylation/dephosphorylation of several intracellular proteins involved in the MAPK pathway. Moreover, MKIs target growth factor receptors such as the vascular endothelial growth factor receptor (VEGFR) and the fibroblast growth factor receptor (FGFR), blocking angiogenesis and cell proliferation [2].

To date, several MKIs have been approved by the Food and Drug Administration (FDA) and European Medical Agency (EMA) for radioiodine (RAI) refractory differentiated thyroid cancer (sorafenib and lenvatinib) and medullary thyroid cancer (vandetanib and cabozantinib), while dabrafenib/trametinib combination has obtained regulatory approval from the FDA for anaplastic thyroid cancer with a BRAF V600 mutation [3,4,5,6,7]. Others kinase inhibitors, selpercatinib and pralsetinib, have recently been approved by the FDA for locally advanced or metastatic tumors with a rearranged during transfection (RET) gene fusion [8,9]. In these studies, it has been demonstrated that MKIs significantly improve the progression-free survival (PFS) of patients with advanced disease. In the SELECT trial, patients with progressive locally advanced or metastatic differentiated thyroid cancer treated with lenvatinib had a significantly longer PFS than those who received placebo (18.3 vs. 3.6 months, respectively).

Several in vitro studies evaluated the pharmacokinetics of MKIs that are mainly metabolized in the liver by cytochrome P450 (CYP). These studies showed that CYP3A4, stimulated by cytochrome b5, is mainly involved in MKI metabolism and oxidation, but other cytochromes (i.e., CYP1A1, CYP1B1, CYP3A5, CYP2D6) also seem to be involved in this process [10,11,12,13].

Recent studies demonstrated that there is large inter-individual pharmacokinetic variability among patients treated with MKIs and that MKIs’ steady state is associated with the genetic polymorphisms in the genes coding for these proteins, which influence the absorption and elimination of the drug. In addition, the rate of development of adverse events during the treatment as well as the drug efficacy are significantly associated with these genetic polymorphisms in different human tumors [14,15,16]. A Japanese study evaluated the effect of *CYP3A4-5* and ATP-binding cassette transporter (*ABCC2*) polymorphisms on lenvatinib pharmacokinetics in patients with thyroid cancer. A correlation between lenvatinib steady-state concentration and these polymorphisms was found, but the incidence of some adverse events (AE) (i.e., hypertension, hand foot syndrome, proteinuria), were not related to the presence of cytochrome polymorphism(s) [17]. Conclusive data are still missing in the literature. In this light, we retrospectively evaluated seven variants belonging to cytochrome P450 (*CYP3A4* rs2242480 and rs2687116 and *CYP3A5* rs776746) genes and to ATP-binding cassette transporter (*ABCB1* rs1045642, rs2032582 and rs2235048 and *ABCG2* rs2231142) genes in a small group of advanced thyroid cancer patients treated with lenvatinib to study the possible correlation between polymorphisms (SNPs), outcome and the appearance of drug-related AEs.

## 2. Results

### 2.1. Patients

In this retrospective study, we enrolled 18 advanced thyroid cancer patients (F:11) treated with lenvatinib. The mean age at the time of MKI treatment was 67.5 ± 13.8 years (median 69 years, 30–96 years). At histology, eight patients had differentiated thyroid cancer (DTC), eight patients had a poorly differentiated thyroid cancer (PDTC) and two medullary thyroid cancer (MTC). These last patients were included because in real-life clinical experience, lenvatinib showed interesting results as salvage therapy in patients with advanced progressive metastatic MTC (20). The majority of patients (13/18, 72.3%) presented with distant metastasis in different sites before starting treatment (ranging from one to five different organs involved) and among them, 6/13 had bone lesions. In total, 5/18 (27.7%) had locally advanced disease. Initial daily lenvatinib dose was 24 mg for 14/18 (77.7%) patients and 14 mg for 4/18 (22.3%) patients, due to older age and comorbidities. The dose was adjusted according to clinical conditions or the occurrence of AEs during the follow-up. The mean best-tolerated dose was 19.4 ± 5 mg (median 20 mg, 10–24). Lenvatinib was the first line of treatment for 14/18 (77.8%) patients and the second/third line of treatment for 4/18 (22.2%) patients (Table 1). Only patients without chronic use of concomitant medications that are known to be strong inducers or inhibitors of cytochrome P450 or substrates of the ATP-binding cassette transporters were included. We also excluded patients with severe comorbidities (i.e., gastrointestinal disease) to avoid a possible bias in terms of the evaluation of the adverse events. Considering the already described nine groups of adverse events (AEs), we divided patients according to the occurrence of less than 5/9 AEs or more than 5/9 of AEs, and we did not find any significant difference according to age (*p* = 0.53) or gender (*p* = 0.58). Even evaluating the higher grade of each adverse event and grouping patients according to the occurrence of mainly G1-AEs or G2/G3-AEs, we did not find any significant difference according to age (*p* = 0.93) or gender (*p* = 0.26). We report the observed drug-related adverse events in Table 2.

The mean time of treatment was 23.0 months (range 2.7–52.9 months). During the first 12 months of treatment, 72.3% of patients needed to stop lenvatinib therapy for a mean of 13.5 days (range 5–51) due to the severity of the side effects. During this period, the mean daily dosage in the cohort of patients was 17 ± 5 mg (not assessable in one patient due to lack of data). Overall, the rates of patients who suspended lenvatinib administration and had a subsequent dose reduction after the occurrence of each adverse events were: 50% for diarrhea; 22% for nausea, vomiting and epigastric pain; 0% for oral mucositis and xerostomia; 10% for hypertension; 40% for proteinuria; 60% for asthenia; 33% for anorexia; 0% for hand foot syndrome and weight loss. The median duration of dose reduction was 12 days for diarrhea; 10 days for nausea, vomiting and epigastric pain; 9 days for hypertension; 22.5 days for proteinuria; 8.66 days for asthenia and 15 days for anorexia. The reduction rate of starting dose was 24% for diarrhea; 22% for nausea, vomiting and epigastric pain; 16% for hypertension; 41.6% for proteinuria; 24.8% for asthenia and 28.9% for anorexia. In total, 2/6 patients who experienced asthenia, 1/5 patients who experienced diarrhea and 1/3 patients who experienced anorexia needed a further 30% reduction in drug dosage.

### 2.2. Correlation between SNPs and Side Effects

Table 3 shows selected SNPs with indications of the most common allele in the European population according to the Ensembl database (https://www.ensembl.org/index.html (accessed on 27 January 2023)) and the observed allele frequency/genotypes in our population. Distribution for all SNPs was consistent with Hardy–Weinberg’s law at the level of significance of 0.05. For each variant, we evaluated the correlation with specific side effects in terms of presence/absence, grade and extent both at genotype and allele level.

#### 2.2.1. Diarrhea

Out of 18 analyzed patients, 14 presented with diarrhea (77.8%). We found that both at genotype and allele level, none of the analyzed SNPs influenced the presence/absence of the side effect (Table 4). Patients were then stratified according to the grade of diarrhea into grade 1 (low, n = 9), grade 2 (moderate, n = 3) and grade 3 (severe, n = 2). At genotype level (Table 4), we found a statistically significant (*p* = 0.03) correlation for *CYP3A4* rs2242480 and *CYP3A5* rs776746. For *CYP3A4* rs2242480, the GG genotype and for *CYP3A5* rs776746, the CC genotype were associated with grade 1 (64.5% of the patients for both SNPs). At allele level, these data were confirmed (Table 4) with the G or the C allele mostly associated with the mildest manifestation of the side effect. Accordingly, for *CYP3A4* rs2242480, the GA or AA genotypes were associated with grade 3 and 2 in 100% of the cases. Similar results were observed for *CYP3A5* rs776746. The TT genotype was associated with grade 2 in all patients and the CT genotype with grade 3. We then wondered whether a specific genetic profile was correlated with days of persistence of the maximum grade of the side effect. We stratified our patients accordingly and obtained three groups: less than 1 month (<1), between 1 and 4 months (1–4) and more than 4 months (>4). At genotype level, we did not find any association. Again, for *CYP3A4* rs2242480, the G allele (*p* = 0.015) and for *CYP3A5* rs776746, the C allele (*p* = 0.018) were statistically associated with >4 months (Table 5).

#### 2.2.2. Nausea, Vomiting and Epigastric Pain

Sixty-six percent (12/18; 66.7%) of patients suffered from nausea, vomiting and epigastric pain. In accordance with what was observed for diarrhea, none of the SNPs were associated with the presence/absence of the symptoms (considered together), both at genotype and allele level (Table 6). The majority of patients (75%) who presented with the side effects were grade 1. Only three patients (25%) were grade 2 and none were grade 3. At genotype level (Table 6), although not significant, we found the CC (rs1045642) or TT (rs2235048) genotypes in the *ABCB1* gene to be slightly correlated with grade 1 (*p* = 0.08 and *p* = 0.06, respectively). The T allele in rs2235048 was statistically (*p* = 0.019) more present in patients with grade 1, as indicated in Table 6. Since these SNPs were close to significance, the haplotype was considered, but even combined together, they were not correlated with the grade of side effect (*p* = 0.13). No correlation was found between SNPs and days of persistence of the higher grade of the AE effect, both at genotype and allele levels.

#### 2.2.3. Oral Mucositis and Xerostomia

Several patients (15/18; 83.4%) presented with mucositis and xerostomia: 11/15 (73.4%) were grade 1, 3/15 (20%) were grade 2 and 1/15 (6.6%) was grade 3. None of the patients had the side effects for less than one month; for 5 patients (33.4%), the duration was 1–4 months, and for 10/15 (66.6%), the extent was more than 4 months. We only found a statistically significant (*p* = 0.039) correlation between rs2231142 in the *ABCG2* gene and the duration of the maximum grade at genotype level (Table 7). For this variant, the CC genotype was almost exclusively represented in the “>4 months” group (9/11 patients) and the CA genotype in the “1–4 months” (3/4 patients). No associations were found at allele level.

#### 2.2.4. Hypertension and Proteinuria

In total, 13/18 patients (72.3%) had hypertension. At genotype level, the rs2242480 in *CYP3A4* (GG) and the rs776746 in *CYP3A5* (CC) were associated (*p* = 0.02) with the presence of the side effect (Table 8). These data were confirmed at allele level (Table 8), with the G and C allele found in 75.8% of the patients with hypertension (*p* = 0.001 and *p* = 0.0013, respectively). In total, 2/13 (15.3%) had grade 1, 7/13 (53.8%) had grade 2 and the remaining 30.9% had grade 3. None of the SNPs were significantly associated with the severity and the duration of the side effect (Table 8). No correlation was found between variants and proteinuria at any levels.

#### 2.2.5. Asthenia

When evaluated for asthenia, we found that 53.8% of the patients presented with grade 1, 30.8% with grade 2 and the remaining 15.4% with grade 3. The CC genotype and, consequently, the C allele for rs2231142 in *ABCG2* were statistically associated (*p* = 0.017 and *p* = 0.028, respectively) with grade 1 (Table 9). At allele level, the G allele for rs2242480 (*CYP3A4*, *p* = 0.02) and the C allele for rs776746 (*CYP3A5*, *p* = 0.022) were also associated with grade 1 of asthenia. These SNPs both alone and combined together were slightly (but not significantly) linked to the severity of asthenia (*p* = 0.08) at genotype level. For the majority of patients (59%), the side effect lasted more than 4 months. However, no statistically significant association was found between SNPs and asthenia duration.

#### 2.2.6. Anorexia and Weight Loss

Approximately all patients (78%) experienced anorexia. This side effect was not linked to genotype at any level.

Before treatment, mean body weight was 73.7 ± 20 kg and mean BMI was 27.2 ± 7 kg/m^2^; 5/18 (27.8%) patients were obese (BMI ≥ 30 kg/m^2^), 4/18 (22.2%) were overweight (BMI 25–29.9 kg/m^2^) and the remaining 50% were normal weight (BMI 18.5–24.9 kg/m^2^). During the first year of treatment, the minimum mean weight reached was 66.9 ± 15 kg, ranging from 43.7 to 108.5 kg. Weight loss was evident in 83.4% of patients, and it was ≥5% in 11/18 (61.1%) patients. In our cohort of patients, weight loss was not always present at the same time as anorexia, as it could be associated with diarrhea, nausea or vomiting. In our patients, weight loss was grade 1 in 36.4%, grade 2 in 45.4% and grade 3 in 18.2% of patients. The mean time needed to reach the minimum weight was 10.3 months (2.7–12 months). Both at genotype and allele levels, the *ABCB1* gene was associated with weight loss (Table 10). Specifically, we found a correlation between the heterozygous CT genotype for both rs1045642 and rs2235048 (*p* = 0.003) corresponding to the C (*p* = 0.014) and the T (*p* = 0.02) allele, respectively. The analyzed SNPs were not associated with the severity of weight loss and duration of adverse events at either genotype or allele level.

#### 2.2.7. Hand Foot Syndrome

Fifty per cent of the patients (9/18) developed hand foot syndrome (HFS); 4/9 (44.4%) were grade 1 and 5/9 (55.6%) were grade 2. None of the patients reached grade 3. For 66.7% of the subjects, HFS duration was more than 4 months (mean time 8.3 months), and for the remaining 33.3% of patients, HFS duration was 1 to 4 months (mean 1.48 months). We did not find any association between SNPs and HFS at genotype or allele level.

#### 2.2.8. Correlation between SNPs and Response to Lenvatinib Treatment

As mentioned above, the mean time of treatment was 23.0 months (range 2.7–52.9 months). During the first 12 months of treatment, 72.3% of patients needed to stop lenvatinib consumption for a mean of 13.5 days (range 5–51) due to the severity of the side effects. During this period, the mean daily dosage in the cohort of patients was 17 ± 5 mg (not assessable in one patient due to lack of data). None of the SNPs were significantly associated with the length of therapy interruption at genotype or allele levels. In terms of best response (calculated from the beginning of the treatment to the last CT scan), we observed 50% stable response (SD), 44.5% partial response (PR) and 5.5% progressive disease (PD). Again, no association between SNPs and best response was observed (Table 11). A mean time of 8.83 months (range 2.767–21.233) was necessary to reach the best response.

By grouping this interval into four categories (<5; 5–10; 10–20; >20 months) and excluding patients with a PD, we found an association (*p* = 0.013) with SNPs rs2242480 (*CYP3A4*) and rs776746 (*CYP3A5*). For both variants, heterozygous and rare homozygous patients fall exclusively into the category >20 months (Table 11) at genotype level. This result was even more evident when SNPs were analyzed at allele level (Table 11). Indeed, in both cases, the rare alleles (A for rs2242480, 8% in the general population; T for rs776746, 6% in the general population) significantly (*p* < 0.0001) correlated with the worst time interval.

In order to explore if these results might be due to the lower dose of lenvatinib rather than the presence of SNPS, a comparison between patients with SNPS and patients carrying the most common genotypes was made. A higher dose of lenvatinib was administrated in patients with these SNPs (22.5 mg/die) compared to other patients (16.4 mg/die). In support of this hypothesis, a positive correlation was found between the mean dose of lenvatinib and the time necessary to reach the best response (*p* = 0.009, R = 0.60). Indeed, only 33.3% of patients treated with more than 20 mg/die reached the BR during the first year of treatment compared to 81.8% of patients taking a dosage less than 20 mg/die.

## 3. Discussion

From a chemical point of view, lenvatinib is a member of the class of quinolines and is the carboxamide of 4-{3-chloro-4-[(cyclopropylcarbamoyl) amino] phenoxy}-7-methoxyquinoline-6-carboxylic acid, which works as a multi-kinase inhibitor and antineoplastic agent by targeting vascular endothelial growth factor receptors (VEGFR1–3), fibroblast growth factor receptors (FGFR1–4), platelet-derived growth factor receptor-alpha (PDGFRα), mast/stem cell growth factor receptor (KIT) and rearranged during transfection receptor (RET) proto-oncogenes.

Lenvatinib is orally administered once daily at a maximum dosage of 24 mg in RAI-refractory DTC or at lower dosages according to patient clinical conditions. After oral administration and absorption, lenvatinib binds to plasma proteins (98–99%), mainly albumin, with an elimination t1/2 of 28 h [12]. It is then mainly metabolized in the liver by cytochrome P450, mostly through CYP3A4 (>80%), and also by CYP3A5, which has similar catalytic specificities [17,18]. Lenvatinib metabolites have low pharmacological activity and are excreted mainly via the biliary route [19,20,21]. Drug bioavailability is also determined by the P-glycoprotein encoded by the *ABCB1* gene and the breast cancer resistance protein (BCRP) encoded by the *ABCG2* gene, which are expressed in the small intestine, liver, kidney and blood–brain barrier, and are associated with drug availability, functioning to regulate the absorption and elimination of substrate drugs [22]. Inducers or inhibitors of these proteins can modify drug bioavailability, increasing or decreasing plasma concentration, respectively. For example, the co-administration of ketoconazole, which is a potent CYP3A, P-glycoprotein, and BCRP inhibitor, has been reported to increase the maximum plasma concentration of lenvatinib compared with placebo [23,24].

Patients treated with lenvatinib develop several adverse events such as hypertension, weight loss, anorexia, fatigue and gastrointestinal side effects [6]. Although these AEs are not life-threatening, they can be so severe as to dramatically impair patients’ quality of life with the consequence of a permanent drug withdrawal in a significant number of cases [1,6]. A recent study (17) tried to demonstrate a possible correlation between polymorphisms in the *CYP3A4*, *CYP3A5*, *ABCB1*, *ABCC2* and *ABCG2* genes and the incidence of adverse events during lenvatinib treatment. The authors showed that patients with *CYP3A4* 20230G>A polymorphism exhibited significantly lower dose-adjusted C0 values for lenvatinib. Nevertheless, the incidence of adverse events following lenvatinib treatment appeared not to be related to the investigated polymorphisms [17].

In our pilot study, we aimed to correlate the most common adverse reactions induced by lenvatinib with specific SNPs. We observed that at genotype level, the rs2242480 in *CYP3A4* (GG) and the rs776746 in *CYP3A5* (CC) were significantly associated with the presence of hypertension (*p* = 0.02). The most common genotypes for rs2242480 (*CYP3A4*: GG) and rs776746 (*CYP3A5*: CC) were associated with the mildest grade of diarrhea (64.5% of the patients for both SNPs). For all these SNPs, heterozygous or rare homozygous genotypes were associated with grade 2 or 3 of the side effects in 100% of the cases. Regarding diarrhea, the same genetic profile also correlated with days of persistence of the maximum grade of the side effect (>4 months). A link with weight loss was also observed for heterozygous patients for the *ABCB1* gene, both for rs1045642 and rs2235048.

The *ABCG2* gene was significantly correlated with the duration of grade maximum for mucositis and xerostomia. In this case, the heterozygous CA genotype was linked to shorter duration, whereas the CC genotype linked to a longer extent (>4 months).

Polymorphisms in both cytochrome P450 genes and ATP-binding cassette transporters appear to be involved with asthenia.

From this analysis, even though the group of patients was small, it seems that a particular genetic make-up can predict the occurrence and severity of common side effects that are MKI-related (Figure 1). The most implicated genes were *CYP3A4* and *CYP3A5*. Our results showed that heterozygous and rare homozygous patients for these genes (rs2242480 and rs776746, respectively), took more than 20 months to reach the best response to treatment, compared to patients carrying the most common genotypes. In our cohort, this effect does not seem to be linked to lower drug dose, given that a higher dose of lenvatinib was administrated in patients with these SNPs (22.5 mg/die) compared to other patients (16.4 mg/die). In support of this hypothesis, a positive correlation was found between the mean dose of lenvatinib and the time necessary to reach the best response (*p* = 0.009, R = 0.60). Indeed only 33.3% of patients treated with more than 20 mg/die reached the BR during the first year of treatment, compared to 81.8% of patients taking a dosage less than 20 mg/die.

We are aware that the limited number of patients strongly reduces the power of statistical analysis, and more patients are needed to confirm these data. In addition, since this is a retrospective study, the measurement of lenvatinib plasma concentrations was not available. However, in the era of precision medicine, the results obtained for some SNPs prompt us to suggest a genetic evaluation before starting MKI treatment. In our cohort, for example, patients who experienced diarrhea and asthenia required a suspension or a significant reduction in the dosage up to the half-life of the starting dose. With this in mind, predicting the occurrence and grade of some side effects will contribute to improving clinical management during lenvatinib treatment in patients with advanced thyroid carcinoma.

## 4. Materials and Methods

### 4.1. Patients

This study retrospectively analyzed 18 patients with advanced thyroid cancer treated with lenvatinib, followed at the Unit of Endocrinology of Siena’s University Hospital (Italy), between November 2004 and May 2022. Informed consent has been signed by each patient enrolled in the study, and the study was approved by our local Ethical Committee (Ethics Committee of Region Toscana, Area Vasta Sud Est, AOUS. Protocol ID: 10167). The data collected included age at diagnosis, gender, anthropometric parameters, histological findings, stage at diagnosis, numbers of anatomical sites involved, tumor response and data on last follow-up/death. At baseline and during periodic follow-up, information regarding lenvatinib treatment was also collected: time lapse between diagnosis and treatment start, mean daily dosage, days of treatment suspension/reduction, appearance, grade and duration of AEs and duration of treatment. We evaluated the occurrence of the more common lenvatinib AEs during the first year of treatment or at last follow-up (if treatment lasted less than one year), grouped as follows: (1) diarrhea; (2) nausea, vomiting and epigastric pain; (3) oral mucositis and xerostomia; (4) hypertension and proteinuria; (5) asthenia; (6) anorexia and weight loss; (7) hand foot syndrome. For each group of AEs, we considered the overall grade and duration of the AE with the highest grade and duration.

The severity of AEs was defined according to the NCI Common Terminology Criteria for Adverse Events v5.0 [https://ctep.cancer.gov/protocolDevelopment/electronic_applications/ctc.htm (accessed on March 2022)].

Radiological evaluation was performed at baseline (before starting lenvatinib) and periodically (on average, at 3 and 6 months and thereafter annually) with contrast-enhanced computed tomography (CT) scanning. Treatment response was classified according to the Response Evaluation Criteria in Solid Tumors (RECIST) v.1.1 (18).

### 4.2. DNA Extraction and SNPs Analysis

Genomic DNA was extracted from peripheral blood leukocytes using the salting out procedure, obtained by fasting venous blood samples collected during lenvatinib treatment. DNA concentration and purity were assessed with a Nanodrop One (Thermo Scientific, Milan, Italy). The analysis of SNPs (Table 1) was carried out through PCR amplification. Specific primers were designed using the Primer3 Input program and purchased from Eurofins Genomics (Ebersberg, Germany). Primer sequences and PCR conditions are reported in the Appendix A. PCR products were subjected to Sanger sequencing using the BigDye Terminator v1.1 Cycle Sequencing Kit (Applied Biosystems, Milan, Italy) and BigDye Xterminator Purification Kit on an automated DNA capillary sequencer (Applied Biosystems 3130xl Genetic Analyzer). Electropherograms were visualized using Chromas (http://technelysium.com.au/wp/chromas/ (accessed on March 2022)) and aligned with reference sequences using the Expasy SIM-Alignment Tool (https://web.expasy.org/sim/ (accessed on March 2022)).

### 4.3. Statistical Analysis

All statistical analyses were carried out by using the software Statview version 5.0.1 for Windows (SAS Institute Inc., Cary, NC, USA). A *p*-value < 0.05 was considered statistically significant.

The Hardy–Weinberg equilibrium was evaluated using the online calculator Gene Calc (https://gene-calc.pl/hardy-weinberg-page (accessed on 1 April 2022)). Haplotype frequencies and association statistics for the polymorphisms were evaluated using Haploview (19).

Association analyses were carried out using *χ*2 or Fischer’s exact tests at genotype, allele and haplotype levels. At allele level, the additive model was considered.

## Figures and Tables

**Figure 1 ijms-24-05496-f001:**
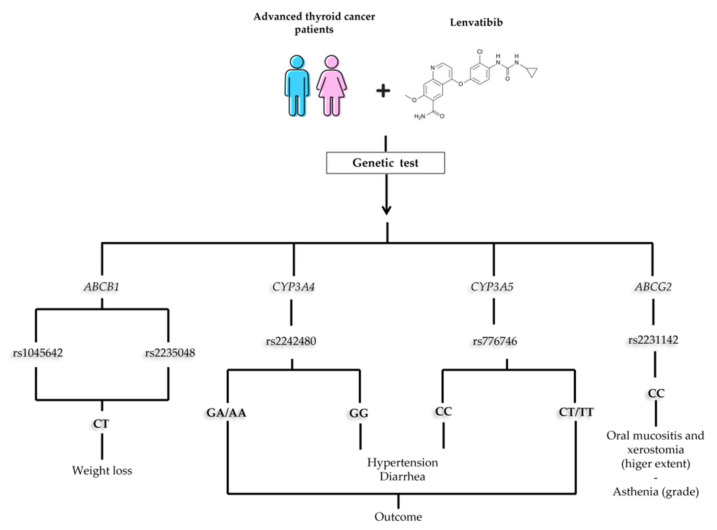
Suggested flowchart of genetic analysis at the beginning of therapy with lenvatininb.

**Table 1 ijms-24-05496-t001:** Clinical characteristics of patients.

Characteristics of Patients, n = 18	
Age at TKI start, years	
Median (range)	69 (30–96)
Mean ± SD	67.5 ± 13.8
Gender, n (%)	
Female	11 (61)
Male	7 (39)
Histology, n (%)	
DTC	8 (44.5)
PDTC	8 (44.5)
MTC	2 (11)
Site of metastasis, n (%)	
Distant metastasis	13 (72.3)
Local disease	5 (27.7)
Patients with of bone metastasis, n (%)	6 (33.3)
Numbers of anatomical site involved, n (%)	
1	5 (27.8)
2	4 (22.2)
≥3	9 (50)
Initial dosage, n (%)	
24 mg	14 (77.7)
14 mg	4 (22.3)
Lenvatinib line of treatment, n (%)	
First	14 (77.8)
Second/third	2 (22.2)
Best response	
PD	1 (5.5)
PR	8 (44.5)
SD	9 (50)
Time to reach the best response, months	
Median (range)	7.4 (1.9–21.2)
Mean ± SD	8.8 ± 6
Duration of lenvatinib treatment, months	
Median (range)	25.9 (2.7–52.9)
Mean ± SD	23.0 ± 15.1

**Table 2 ijms-24-05496-t002:** Prevalence, grade and duration of the adverse events.

Adverse Event	AE Presence n (%)	Grade 1 n (%)	Grade 2 n (%)	Grade 3 n (%)	Duration of the AE (Months) Mean ± SD Median (Range)	Duration of the G Max AE (Months) Mean ± SD Median (Range)
**Diarrhea**	14 (77.8)	9 (64.3)	3 (21.4)	2 (14.3)	5.8 ± 3.7 5.2 (0.3–12)	4.7 ± 4.4 4.2 (0.06–12)
**Nausea, vomiting and epigastric pain**	12 (66.7)	9 (75)	3 (25)	0 (0)	4.5 ± 4.8 2.5 (0.03–12)	4.2 ± 4.8 1.9 (0.03–11.9)
**Oral mucositis and xerostomia**	15 (83.4)	11 (73.4)	3 (20)	1 (6.6)	7.5 ± 4.4 8.6 (1–12)	6.0 ± 4.5 4.2 (1–12)
**Hypertension**	13 (72.3)	2 (15.3)	7 (53.8)	4 (30.9)	9.2 ± 4.2 30 (2.6–12)	9.1 ± 3.9 30 (2.6–12)
**Proteinuria**	5 (29.4)	2 (40)	1 (20)	2 (40)	4.9 ± 4.3 2 (1.3–9.7)	1.24 ± 0.7 1.3 (0.33–2)
**Asthenia**	13 (76.5)	7 (53.8)	4 (30.8)	2 (15.4)	7.6 ± 4.3 7.3 (1.4–12)	5.5 ± 4.9 4.6 (0.13–12)
**Anorexia**	13 (76.5)	7 (53.8)	2 (15.4)	4 (30.8)	7.9 ± 4.7 10.3 (0.6–12)	5.6 ± 5.1 3.8 (0.13–12)
**Hand-foot syndrome**	9 (50)	4 (44.4)	5 (55.6)	0 (0)	6.6 ± 4.3 5.8 (1.3–12)	6.0 ± 4.3 4.5 (1.3–12)
**Weight loss**	11 (61.1)	4 (36.4)	5 (45.4)	2 (18.2)	*	*

* Mean time to reach the minimum weight 10.3 ± 2.6 months, median 11.6 (range 4.96–12.3) months.

**Table 3 ijms-24-05496-t003:** SNPs analyzed in the cohort of patients with advanced thyroid carcinoma.

Gene	Polymorphism	Type of SNP (Ensembl Database)	Observed Allele Frequency in Our Cohort (%)	Allele Frequency in the European Population (%)	Genotype (n = 18)
** *CYP3A4* **	rs2687116	Intronic	A: 94.4 C: 5.6	A: 97 C: 3	AA: 16 AC: 2 CC: 0
	rs2242480	Intronic	G: 91.7 A: 8.3	G: 92 A:8	GG: 16 GA: 1 AA: 1
** *CYP3A5* **	rs776746	Splice acceptor variant	C: 86.2 T: 13.8	C: 94 T:6	CC: 15 CT: 1 TT: 2
** *ABCB1* **	rs2032582	p.S893A	G: 55.5 T: 44.5	G: 59 T: 41	GG: 4 GT: 12 TT: 2
	rs1045642	p.Ile1145Met	T: 41.6 C: 58.4	T: 52 C: 48	CC: 5 CT: 11 TT: 2
	rs2235048	Intronic	C: 38.9 T: 61.1	C: 52 T: 48	CC: 2 CT: 10 TT: 6
** *ABCG2* **	rs2231142	p.Gln141Glu	C: 88.9 A: 11.1	C: 91 A: 9	CC: 14 CA: 4 AA: 0

The results are detailed below.

**Table 4 ijms-24-05496-t004:** *p* value by chi-squared test at genotype and allele level for the presence/absence and the grade of diarrhea. Correlation between diarrhea side effect and polymorphisms linked to MKI metabolism.

		Presence/Absence of Diarrhoea		Grade of Diarrhoea	
Gene	Polymorphism	*p* Value (Genotype Level)	*p* Value (Allele Level)	*p* Value (Genotype Level)	*p* Value (Allele Level)
** *CYP3A4* **	rs2687116	0.42	0.43	0.26	0.29
** *CYP3A4* **	rs2242480	0.64	0.3	0.03 *	0.04 *
** *CYP3A5* **	rs776746	0.64	0.33	0.03 *	0.04 *
** *ABCB1* **	rs2032582	0.27	0.72	0.72	0.71
** *ABCB1* **	rs1045642	0.28	0.17	0.37	0.67
** *ABCB1* **	rs2235048	0.22	0.12	0.11	0.24
** *ABCG2* **	rs2231142	0.08	0.89	0.4	0.45

* statistically significant.

**Table 5 ijms-24-05496-t005:** Association between the G allele (*CYP3A4* rs2242480) and the C allele (*CYP3A5* rs776746) with months of persistence of the maximum grade of diarrhea.

	Diarrhea (Months)			
Allele	<1	1–4	>4	*p*
*CYP3A4* rs2242480				
**A**	3	0	0	0.015 *
**G**	5	6	14	
*CYP3A5* rs776746				
**C**	5	6	14	0.018 *
**T**	3	0	0	

* statistically significant.

**Table 6 ijms-24-05496-t006:** *p* value by chi-squared test at genotype and allele level for the presence/absence and the grade of nausea, vomiting and epigastric pain. Correlation between nausea, vomiting and epigastric pain side effects and polymorphisms linked to MKI metabolism.

		Presence/Absence of Nausea, Vomiting and Epigastric Pain		Grade of Nausea, Vomiting and Epigastric Pain	
Gene	Polymorphism	*p* Value (Genotype Level)	*p* Value (Allele Level)	*p* Value (Genotype Level)	*p* Value (Allele Level)
** *CYP3A4* **	rs2687116	0.28	0.29	0.37	0.42
** *CYP3A4* **	rs2242480	0.56	0.19	0.67	0.3
** *CYP3A5* **	rs776746	0.56	0.19	0.67	0.28
** *ABCB1* **	rs2032582	0.37	0.71	0.13	0.16
** *ABCB1* **	rs1045642	0.17	0.18	0.08	0.1
** *ABCB1* **	rs2235048	0.1	0.12	0.06	0.019 *
** *ABCG2* **	rs2231142	0.68	0.67	0.7	0.72

* statistically significant.

**Table 7 ijms-24-05496-t007:** Genotype associated with the duration of the maximum grade of mucositis and xerostomia at genotype level for rs2231142 in *ABCG2* gene. *p* < 0.05 with *χ*2.

	Oral Mucositis and Xerostomia (Months)		
Genotype	1–4	>4	*p*
*ABCG2* rs2231142			
**CC**	2	9	0.039 *
**CA**	3	1	

* statistically significant.

**Table 8 ijms-24-05496-t008:** *p* value by chi-squared test at genotype and allele level for the presence/absence and grade of hypertension. Correlation between hypertension side effect and polymorphisms linked to MKI metabolism.

		Presence/Absence of Hypertension		Grade of Hypertension	
Gene	Polymorphism	*p* Value (Genotype Level)	*p* Value (Allele Level)	*p* Value (Genotype Level)	*p* Value (Allele Level)
** *CYP3A4* **	rs2687116	0.35	0.38	0.68	0.64
** *CYP3A4* **	rs2242480	0.02 *	0.001 *	0.54	0.39
** *CYP3A5* **	rs776746	0.02 *	0.0013 *	0.91	0.8
** *ABCB1* **	rs2032582	0.61	0.76	0.5	0.47
** *ABCB1* **	rs1045642	0.27	0.74	0.08	0.25
** *ABCB1* **	rs2235048	0.27	0.74	0.09	0.25
** *ABCG2* **	rs2231142	0.65	0.7	0.27	0.28

* statistically significant.

**Table 9 ijms-24-05496-t009:** *p* value by chi-squared test at genotype and allele level for the presence/absence and grade of asthenia. Correlation between asthenia side effect and polymorphisms linked to MKI metabolism.

		Presence/Absence of Asthenia		Grade of Asthenia	
Gene	Polymorphism	*p* Value (Genotype Level)	*p* Value (Allele Level)	*p* Value (Genotype Level)	*p* Value (Allele Level)
** *CYP3A4* **	rs2687116	0.4	0.9	0.72	0.74
** *CYP3A4* **	rs2242480	0.7	0.56	0.08	0.02 *
** *CYP3A5* **	rs776746	0.7	0.56	0.08	0.022 *
** *ABCB1* **	rs2032582	0.24	0.4	0.14	0.11
** *ABCB1* **	rs1045642	0.64	0.4 *	0.28	0.49
** *ABCB1* **	rs2235048	0.64	0.82	0.28	0.49
** *ABCG2* **	rs2231142	0.28	0.9	0.017 *	0.028 *

* statistically significant.

**Table 10 ijms-24-05496-t010:** *p* value by chi-squared test at genotype and allele level for weight loss. Correlation between weight loss side effect and polymorphisms linked to MKI metabolism.

		Presence/Absence of Weight Loss		Grade of Weight Loss	
Gene	Polymorphism	*p* Value (Genotype Level)	*p* Value (Allele Level)	*p* Value (Genotype Level)	*p* Value (Allele Level)
** *CYP3A4* **	rs2687116	0.5	0.49	0.28	0.2
** *CYP3A4* **	rs2242480	0.79	0.39	0.17	0.15
** *CYP3A5* **	rs776746	0.79	0.39	0.17	0.15
** *ABCB1* **	rs2032582	0.3	0.18	0.18	0.94
** *ABCB1* **	rs1045642	0.003 *	0.014 *	0.8	0.49
** *ABCB1* **	rs2235048	0.003 *	0.02 *	0.8	0.29
** *ABCG2* **	rs2231142	0.61	0.22	0.06	0.14

* statistically significant.

**Table 11 ijms-24-05496-t011:** *p* value by chi-squared test at genotype and allele level for best response and interval to reach the best response. Correlation between response to Lenvatinib and polymorphisms linked to MKI metabolism.

		Best Response		Time Interval	
Gene	Polymorphism	*p* Value (Genotype Level)	*p* Value (Allele Level)	*p* Value (Genotype Level)	*p* Value (Allele Level)
** *CYP3A4* **	rs2687116	0.14	0.97	0.17	0.26
** *CYP3A4* **	rs2242480	0.68	0.87	0.013 *	<0.0001 *
** *ABCB1* **	rs2032582	0.24	0.57	0.14	0.6
** *ABCB1* **	rs1045642	0.5	0.66	0.11	0.08
** *ABCB1* **	rs2235048	0.3	0.43	0.11	0.07
** *ABCG2* **	rs2231142	0.13	0.31	0.5	0.57
** *CYP3A5* **	rs776746	0.68	0.87	0.013 *	<0.0001 *

* statistically significant.

## Data Availability

The data presented in this study are available on request from the corresponding author. The data are not publicly available due to patients’ privacy.

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
