# Peer review of "Pilot Study on the Impact of Polymorphisms Linked to Multi-Kinase Inhibitor Metabolism on Lenvatinib Side Effects in Patients with Advanced Thyroid Cancer"

_ijms, 2023, doi:10.3390/ijms24065496_

Round 1

Reviewer 1 Report

The authors investigated the correlation between the polymorphisms (SNPs) such as CYP genes and ABC transporters genes and the development of AEs in patients with advanced thyroid cancer during the lenvatinib therapy.

Major

1)     There are fundamental flaws in this study. The study lacks measurement of lenvatinib exposures and the number of subjects studied is too small as described in the limited section.

2)     The authors should include the therapeutic details of lenvatinib administration including the onset days of each adverse events, the dose reduction rate, the median duration of dose reduction, and the best tolerated dose.

3)     The substrates of BCRP and/or P-gp might have same effect on activity of these transporters. Were patients who took any substrates of BCRP and/or P-gp excluded?

4)     The authors investigated the association between the SNPs of CYP and BCRP with duration of development of diarrhea. What is the mechanism of the development of diarrhea during the lenvatinib therapy.

5)     The authors state that weight loss was not necessarily linked to anorexia. Further discussion needs to be included to the relationship between anorexia and weight loss.

6)     There are a large number of grammatical errors of English.

Author Response

We thank the reviewer for the useful comments which have been included in the revised version of the manuscript. All changes are highlighted in yellow.

  • There are fundamental flaws in this study. The study lacks measurement of lenvatinib exposures and the number of subjects studied is too small as described in the limited section.

Answer: We agree with the referee but unfortunately, given the retrospective design of the study, these data were not available. We hope to implement our results with a prospective study in which these additional evaluations will be performed. Indeed, this limitation has been already highlighted in the discussion. To stress the nature of the study we change the title in "Pilot study on the impact of polymorphisms linked to multikinase inhibitors metabolism on lenvatinib’s side effects in patients with advanced thyroid cancer".

  • The authors should include the therapeutic details of lenvatinib administration including the onset days of each adverse events, the dose reduction rate, the median duration of dose reduction, and the best tolerated dose.

Answer: After a comprehensive review of lenvatinib administration, we observed that in our patients each time the drug was suspended, it was subsequently resumed at a reduced dose. Overall, the rates patients who suspended lenvatinib administration and then reduced the dosage after the occurrence of each adverse events were: 50% for Diarrhea, 22% for Nausea, vomiting and epigastric pain, 0% for Oral mucositis and xerostomia, 10% for Hypertension, 40% for Proteinuria, 60% for Asthenia, 33% for Anorexia, 0% for Hand-foot Syndrome and Weight loss. The median duration of dose reduction was 12 days for Diarrhea, 10 days for Nausea, vomiting and epigastric pain, 9 days for Hypertension, 22.5 days for Proteinuria, 8.66 days for Asthenia and 15 days for Anorexia. The reduction rate of starting dose was 24% for Diarrhea, 22% for Nausea, vomiting and epigastric pain, 16% for Hypertension, 41.6% for Proteinuria, 24.8% for Asthenia and 28.9% for Anorexia. Two/6 patients who experienced Asthenia, 1/5 patients who experienced Diarrhea and 1/3 patients who experienced Anorexia needed further reduction of 30% of drug dosage. The mean best tolerated dose was 19.4±5 mg (median 20 mg, 10-24).

We added these informations after table 2, in the Results-Patients section.

  • The substrates of BCRP and/or P-gp might have same effect on activity of these transporters. Were patients who took any substrates of BCRP and/or P-gp excluded?

Answer: In the study we selected only patients without chronic use of concomitant medication that are known to be strong inducer or inhibitors of cytochrome P450 or substrate of the ATP-binding cassettes. We reported and better stress this point in the text (“Results" section, 2.1. Patients, page 2).

  • The authors investigated the association between the SNPs of CYP and BCRP with duration of development of diarrhea. What is the mechanism of the development of diarrhea during the lenvatinib therapy.

Answer: Diarrhea was a very common AE in lenvatinib SELECT study arm (about 65% of the patients) (Schlumberger M, Tahara M, Wirth LJ, et al. N Engl J Med. 2015) and a recent meta-analysis of phase 2 and 3 clinical trials, showed that gastrointestinal (GI) AEs are a common class side effect of VEGFR-TKIs (Li J, Gu J. Eur J Clin Pharmacol. 2017). However, the pathophysiology of VEGF inhibitor–induced diarrhea is not fully understood. The proposed mechanisms were the local irritation by metabolites in the feces and transient lactose intolerance, the target effects on kinases within the gut (c-KIT) and the inhibition of microcirculation in the GI tract (Bowen JM. Curr Opin Support Palliat Care. 2013). The excess chloride secretion, which causes a secretive form of diarrhea, was also postulated (Pessi MA, et al. Crit Rev Oncol Hematol.

2014). As reported in our study, gene variants within metabolic pathways for TKIs could also play a role in toxicity susceptibility (Lu JF, Eppler SM, Wolf J, et al. Clin Pharmacol Ther 2006).

  • The authors state that weight loss was not necessarily linked to anorexia. Further discussion needs to be included to the relationship between anorexia and weight loss.

Answer: Weight loss was not always present at the same time with anorexia in our cohort of patient, as clarify in the text (section “Results”, 2.2. Correlation between SNPs and side effects, Anorexia and weight loss, page 7). Indeed, weight loss could occur even without anorexia, as it could be associated with diarrhea, nausea, vomiting, etc. It is not a generalization, but it is what we observed in our group of patients. 

  • There are a large number of grammatical errors of English.

      Answer: English grammar has been revised as suggested.

Reviewer 2 Report

The enclosed article conducted a retrospective study over the advanced thyroid cancer patients receiving respective therpies. By an overview of the genetic backgrounds, the authors intend to suggest certain connections between genotypes and the common side effects from MKI treatments. The rationale is straightforward and clearly stated. Methods and Results were properly drafted; however, some significant setbacks were also noticed.

1) The case number were only 18 and this can't be representative enough. With such limited case number, I wonder how authors correlate all side effects with respective epidemiological items, such as gender and ages. According to the supplementary table 2, none of the items were considered part of the side effect risk factors, particularly the diversified histological categories. The authors should explain this in detail with moving the supplementary table to the main texts with statistical analysis for each item. 

2) With limited case numbers (n=18), I think it's merely justified for the siginificance by p<0.05. For each marginal cases, the authors should declare the difficulty for more case numbers or the ambiguous conclusion may be misleading. 

3) For readers, I think the most important part must be whether MKI is suitable for treating a patient with "severe side effect risks." However, most of the AE mentioned in the article are not really life-threatening or SAE. In this regard, I don't find any particular reason to suggest the patients not to seek MKI treatment, neither the true value of this research. The authors may want to further elaborate the contents in detail in discussion. 

Author Response

We thank the reviewer for the useful comments which have been included in the revised version of the manuscript. All changes are highlighted in yellow.

1) The case number were only 18 and this can't be representative enough. With such limited case number, I wonder how authors correlate all side effects with respective epidemiological items, such as gender and ages. According to the supplementary table 2, none of the items were considered part of the side effect risk factors, particularly the diversified histological categories. The authors should explain this in detail with moving the supplementary table to the main texts with statistical analysis for each item. 

Answer: We agree with the referee but unfortunately, given the retrospective design of the study, the sample size is limited. To stress this fact, we modify article title into " Pilot study on the impact of polymorphisms linked to multi-kinase inhibitors metabolism on lenvatinib’s side effects in pa-tients with advanced thyroid cancer". Moreover, according to our series, all patients have experienced at least two of the adverse events analyzed (diarrhea, nausea, vomiting and epigastric pain; oral mucositis and xerostomia; hypertension and proteinuria; asthenia; anorexia and weight loss; hand-foot syndrome) during Lenvatinib treatment. The majority of them experienced multiple adverse events (median 78% of the AEs).

In the present study we considered 9 groups of adverse events (AEs). By dividing patients according to the occurrence of less than 5/9 AEs or more than 5/9 of AEs, we didn’t find any significant difference according to age (p=0.53) or gender (p=0.58). We have also considered the higher grade of each adverse event and we have grouped patients according to the occurrence of mainly G1-AEs or G2/G3-AEs. Even in this case we didn’t find any significant difference according to age (p=0.93) or gender (p=0.26).

These two statements have been added in the results section, 2.1 patients, page 3.

2) With limited case numbers (n=18), I think it's merely justified for the significance by p<0.05. For each marginal cases, the authors should declare the difficulty for more case numbers or the ambiguous conclusion may be misleading. 

We agree with the referee. In some cases p value was close to statistical significance and probably, a significant association would be highlighted in a larger cohort of patients. Otherwise, we are aware that by increasing the series it is possible that some results may lose statistical significance, therefore we have reformulated the title into "Pilot study on the impact of polymorphisms linked to multikinase inhibitors metabolism on lenvatinib side effects in patients with advanced thyroid cancer".

3) For readers, I think the most important part must be whether MKI is suitable for treating a patient with "severe side effect risks." However, most of the AE mentioned in the article are not really life-threatening or SAE. In this regard, I don't find any particular reason to suggest the patients not to seek MKI treatment, neither the true value of this research. The authors may want to further elaborate the contents in detail in discussion. 

Answer: To date, the systemic therapeutic options for patients with advanced thyroid cancer are limited to MKI and, unfortunately, about 20% of patients must discontinue the treatment because of adverse events (Wells SA, et al. J Clin Oncol 2012; Elisei R, et al. J Clin Oncol 2013, Brose MS, et al. Lancet 2014; Schlumberger M, et al. NEJM 2015; Subbiah V, J Clin Oncol 2018; Cabanillas ME, Endocr Rev 2019). In real life, one of the most important challenge is the management of the adverse event(s) to maintain the therapeutic dosage of MKI as long as possible. In this setting a prior knowledge of possible presentation or grade and duration of the adverse events could help clinicians to manage MKI treatment. We are aware that the results of our study are foreplay and need to be validated in a larger cohort of patients, however they suggest a link between SNP and AEs/prognosis to be taken into consideration and investigated in future studies.

Reviewer 3 Report

This is an interesting and necessary pharmacogenetic study on MKI chemotherapeutics in thyroid cancer. Similar studies in European cohorts have not been published to the best of my knowledge. Although its importance would be greater if the study included analysis of lenvatinib trough levels I assume that this was not achievable due to the long duration of the sample collection as advanced thyroid cancer is not very common.  I also think this justifies the small sample size that the authors are pointing out in the limitations of the study. I have only a few minor comments

1. The last paragraph in the discussion should elaborate a bit on how these results can be employed in the clinic, in terms of personalised medicine. How can genetic information help clinicians design better AE management for each patient? Regarding this matter, I find the flowchart in Figure 3 slightly unclear. How would the genetic test results at the end of the chart be interpreted? For me, the next step, or an explanation in the caption is missing. 

2. Perhaps the correlation of the dosage levels and SNPs with best response (and even AEs that correlate with SNPs) can be included in the results. It is only mentioned in a paragraph in the disscusion, but it might be interesting to present them in the result section as well.

3. Is there a particular reason that Figures 1 and 2 are not presented as tables but rather as figures?

Author Response

We thank the reviewer for the useful comments which have been included in the revised version of the manuscript. All changes are highlighted in yellow.

  1. The last paragraph in the discussion should elaborate a bit on how these results can be employed in the clinic, in terms of personalised medicine. How can genetic information help clinicians design better AE management for each patient? Regarding this matter, I find the flowchart in Figure 3 slightly unclear. How would the genetic test results at the end of the chart be interpreted? For me, the next step, or an explanation in the caption is missing. 

Answer: we provided more details in the discussion and modified the figure with a flowchart.

  1. Perhaps the correlation of the dosage levels and SNPs with best response (and even AEs that correlate with SNPs) can be included in the results. It is only mentioned in a paragraph in the discussion, but it might be interesting to present them in the result section as well.

Answer: We agree with the referee and these data have been included in the results.

  1. Is there a particular reason that Figures 1 and 2 are not presented as tables but rather as figures?

Answer: We agree with the referee and we replaced figures with tables.

Round 2

Reviewer 1 Report

The revised manuscript has been revised well.

Reviewer 2 Report

The authors have addressed all my comments accordingly.